# Do Spiders Ride on the Fear of Scorpions? A Cross-Cultural Eye Tracking Study

**DOI:** 10.3390/ani12243466

**Published:** 2022-12-08

**Authors:** Veronika Rudolfová, Iveta Štolhoferová, Hassan S. A. Elmi, Silvie Rádlová, Kateřina Rexová, Daniel A. Berti, David Král, David Sommer, Eva Landová, Petra Frýdlová, Daniel Frynta

**Affiliations:** 1Department of Zoology, Faculty of Science, Charles University, Viničná 7, 128 43 Prague, Czech Republic; 2Department of Biology, Faculty of Education, Amoud University, Borama, Somaliland

**Keywords:** Africa, arachnophobia, attention, EEA, emotion, fixation, generalization, spontaneous gaze, threat

## Abstract

**Simple Summary:**

In animal phobia research, one of the most attractive topics has been arachnophobia—the specific phobia of spiders. In this study, we explore the apparent paradox of mostly harmless spiders being the object of one of the most common animal fears. Recently, it has been suggested that negative emotions associated with spiders could be triggered by a more generalized fear of chelicerates, where scorpions are the primordial model that one should be afraid of. This hypothesis anticipates that deep fear of scorpions was present in human ancestors and that nowadays it is still generally shared among cultures. To test this assumption, we recruited participants from the Republic of Somaliland and the Czech Republic for an eye-tracking experiment. We found a very strong attentional bias for scorpions as opposed to spiders in Somalis and a similar albeit smaller bias in Czechs. The study deals with possible evolutionary origins of the fear of spiders and arachnophobia, one of the most common specific animal phobias. Moreover, it adds to a very limited number of studies focusing on people’s perception of animals in Sub-Saharan Africa.

**Abstract:**

Deep fear of spiders is common in many countries, yet its origin remains unexplained. In this study, we tested a hypothesis based on recent studies suggesting that fear of spiders might stem from a generalized fear of chelicerates or fear of scorpions. To this end, we conducted an eye tracking experiment using a spontaneous gaze preference paradigm, with spiders and scorpions (previously neglected but crucial stimuli) as threatening stimuli and grasshoppers as control stimuli. In total, 67 participants from Somaliland and 67 participants from the Czech Republic were recruited and presented with a sequence of paired images. Both Somali and Czech people looked longer (total duration of the gaze) and more often (number of fixations) on the threatening stimuli (spiders and scorpions) when presented with a control (grasshopper). When both threatening stimuli were presented together, Somali participants focused significantly more on the scorpion, whereas in Czech participants, the effect was less pronounced, and in Czech women it was not significant. This supports the hypothesis that fear of spiders originated as a generalized fear of scorpions. Moreover, the importance of spiders as fear-eliciting stimuli may be enhanced in the absence of scorpions in the environment.

## 1. Introduction

Humans have a unique attitude towards the environment that surrounds them. They tend to respond strongly and pay special attention to various characteristics that have played crucial roles in the evolution and survival of humankind. In this regard, two concepts are referred to quite often—biophilia and biophobia [1]. Even though these concepts are mostly discussed in the connection with landscapes, they may also be applied to animals [2,3]. Specifically, biophobia represents a disposition to readily associate and later retain a negative response (e.g., fear) or avoidance of stimuli that may pose dangers to the person. It is believed that biophobia is linked with stimuli that have presented danger to survival during the evolution of early humans (predators, snakes, or spiders), and that this fear has remained preserved during evolution [2]. Correspondingly, behavioral responses may be detected even in areas where the stimuli are no longer present [2].

One of the manifestations of biophobia (and biophilia as well) may be, for example, the biologically prepared learning widely discussed by Seligman [4], which is characterized by extraordinarily short reaction times to stimuli that might pose a potential danger to the person. Another intriguing piece of evidence comes from subliminal presentation experiments, which have shown that dangerous stimuli are perceived even though the participant has no recollection of seeing the image [5,6]. Moreover, emotional (or dangerous) stimuli induce attentional biases. These are commonly reported for emotional stimuli—fearful faces and body postures [7,8]—as well as for animals [9,10,11,12] that seem to present a specific category of stimuli with a strong tendency of attracting attention. This fine-tuning of attention for animal stimuli probably had an adaptive value for our hunter–gatherer ancestors. For them, many animals represented either a dangerous predator or a valuable resource, in either case, something worth paying attention to [13].

The most researched animal in the context of visual attention is probably the snake. In her influential work, Lynne Isbell [14] suggested that the evolution of the primate visual system was strongly impacted by the need to rapidly detect fearful stimuli, especially venomous snakes. Her concept came to be known as the Snake Detection Theory, and much additional evidence in favor of this concept was found in the following years [15,16,17,18,19].

It is, however, worth considering whether other life-threatening stimuli might make use of the same brain pathways. Spiders, for example, have been repeatedly shown to evoke great fear, or even phobia, in a wide variety of participants [20,21,22]. Nonetheless, adult participants detected snakes more quickly or accurately than spiders in a visual search task [17,23,24]. Event-related potential studies further supported the unique nature of snakes, in contrast to spiders, in visual processing [25,26], and a similar conclusion was reached by employing the visual search task in Japanese macaques [27], reviewed in [28].

What might be behind the discrepancy between these two stimuli? While both snakes and spiders are considered as similarly fear-evoking animals [29], their actual life-threatening potential is incomparable. From a medical point of view, spiders are considered essentially harmless to humans [30,31,32], whereas snakebites are estimated to be responsible for 81,000–138,000 deaths across the globe annually [33]. This fact has not been lost on the scientific community, and the evolutionary origins of fear of snakes and fear of spiders were repeatedly proposed as not directly comparable [34,35,36].

A recent study on US university students focused on spiders as well as on related and also threatening stimuli, scorpions [37]. Contrary to the authors’ expectations, their results showed a strong correlation of fear of spiders and fear of scorpions, with the fear of scorpions being equal to or even higher than the fear of spiders. In a follow-up study, Frynta and colleagues [36] asked participants to rank an array of live species according to fear, disgust, and beauty. They found that scorpions, spiders, and other chelicerate species were perceived as a unified group of organisms eliciting greater fear and disgust than other tested arthropods (myriapods, hemimetabolous insects, beetles, and crabs). Based on this and a similar study using pictures [38], they proposed a hypothesis concerning the origin of human fear of spiders. They suggest that fear of spiders might be triggered by a generalized fear of chelicerates, where a scorpion is the original stimulus that signals danger. For the extended discussion, see [36]. Indeed, scorpions pose a significant threat to humans, as there are around 1.5 million stings worldwide reported every year, with approximately 2600 of the encounters being lethal [39].

Unfortunately, very little is known about attitudes toward scorpions or spiders across the world, especially in Sub-Saharan Africa. Moreover, the results of the few studies are ambiguous. On the one hand, children from South Africa named spiders among the ten most frightening things [40], and high schoolers from South Africa and Slovakia showed a similar, slightly negative attitude toward spiders [41]. On the other hand, Lemelin and Yen [42] reviewed the available evidence, mainly classic works of anthropologists and ethnobiologists, and concluded that in many cultures, spiders are viewed in a rather positive light. We were not able to find any studies that would allow us to investigate potential cross-cultural differences in attitude toward scorpions.

To address this issue, we conducted an eye tracking experiment utilizing a simple design of spontaneous gaze preference when presented with two stimuli at once. It has been shown that in such a paradigm, fear-eliciting animals are fixated upon more quickly, more frequently, and/or for a longer time period [43,44,45]. Therefore, we argue that this design is well suited for the investigation of unintentional attention bias. To focus on the evolutionary aspect of the introduced hypothesis, we recruited participants in Somaliland and the Czech Republic. The Republic of Somaliland declared independence, but it has remained internationally unrecognized. Somaliland lies in the Horn of Africa, the area traditionally accepted as the cradle of humankind (but see [46,47]). Numerous archaeological sites demonstrate the continuous presence of hominids in this region since at least 4 MY ago [48,49]. Nowadays people of this area belong to the core populations of North and North-East Africa, which have never left the African continent. Ancestors of Somali people have led, or to this day lead nomadic pastoralist lives (this applies to families of university students as well) and have never left the savanna environment. In contrast, the people of the Czech Republic are (just as other Europeans) a derived population that left both the geographic region and the environment of human origin a long time ago. Thus, in this cross-cultural comparison, Somalis represent the probable ancestral condition while deviations can be expected in Czechs.

In our experiment, we predicted the following: (i) Both Somali and Czech participants will pay greater attention to a scorpion and spider when each presented with a control stimulus (a grasshopper; grasshoppers did not elicit fear in [36]). (ii) When both threatening stimuli (scorpion and spider) are presented together, participants will pay greater attention to the scorpion, although the attentional bias will be more pronounced in Somalis. We expected this discrepancy because of differences in encounter rates of these animals during the evolutionary history of Somali and Czech populations. Both populations originated in Africa, where scorpions were (and still are) present; however, scorpions have been missing in the evolutionary environment of the Czech population for the last 4000 years or longer [50]. We, therefore, hypothesize that Czechs would shift part of their attention to spiders, i.e., the secondary stimulus which is nonetheless daily encountered.

## 2. Materials and Methods

### 2.1. Selection and Preparation of the Stimuli

In any eye tracking study, a substantial trade-off must be made between the number of stimuli and the demanding nature of the task [51]. The more stimuli included in the study, the higher generalization possibilities, but on the other hand, the higher the weariness of the participants. For this study, therefore, we chose a set of 36 unique animal pictures in total—12 from each category: spiders, scorpions, and short-horned grasshoppers. Only high-resolution pictures with appropriate licenses were selected (see Appendix A for detailed information about the photo sources). The stimuli were then modified as follows: The background of each animal was cut-off and replaced with uniform 20% grey. Then, the animals were paired into three categories: spider–scorpion, spider–grasshopper, and scorpion–grasshopper. In the first step of pair-matching, the individual pairs were assorted according to the main color, e.g., black animals were matched together, then brown ones, then the ones with disruptive coloration, etc. Next, colors of the animals within a pair were matched (using Adobe Photoshop CS4 v 11.0), i.e., the colors of one animal were transformed to match 50% of the other animal’s original color and vice versa (see Figure 1). This way, we ensured that the two animals in a pair only differed in category-related features (such as shape) and not in color-related features that were not the focus of our study. Using the method described, we created a total number of 18 pairs. Then, we created an additional 18 pairs by mirroring the original pairs horizontally, gaining 36 stimuli pairs in total.

### 2.2. The Experimental Procedure

The participants were seated in front of a computer screen with 1920 × 1080 pixel resolution and asked to watch the presented stimuli during which their eye movements were recorded using the Clinical Eye-Tracker (utilizing the Tobii Bar) by Thomson Software Solutions. The presentation was performed using the Thomson Software Clinical Eye-Tracker Software. Before the presentation began, the eye-tracking camera was calibrated individually for each participant. The stimuli were then presented to each participant in a set in pseudo-random order, in which the stimulus pair and its mirror image had to be interlaced with at least two completely different pairs. Each stimulus pair was presented for 5 s and was interlaced with a black target cross presented for 2 s on the same grey-colored background (Figure 1). To avoid the effect of presentation order, we created another version of the set that was presented in reversed order to half of the participants.

### 2.3. Pilot Experiment

Three months prior to the experiment, we conducted a pilot study with the same design and tested hypotheses in order to check the selected stimuli and estimate the number of participants for the final experimental procedure. A total number of 31 Czech subjects participated in the pilot study. When asked after the experiment, participants reported medium to high fear of spiders and scorpions but only low or no fear of grasshoppers, which thus validated our assumptions about fear elicited by these animals. Analysis of eye movement showed that scorpions were fixated upon more often than grasshoppers; however, the difference was not significant for spider–grasshopper pairs, or spider–scorpion pairs (difference of the number of fixations: scorpion–grasshopper: estimate = 21.29, t = 3.61, *p* < 0.001; spider–grasshopper: estimate = 4.65, t = 0.79, *p* = 0.430; spider–scorpion: estimate = −2.74, t = −0.46, *p* = 0.643). Based on the results, therefore, the grasshopper seemed a suitable control. Following subjective reports of the participants, we color-matched the animal pairs more finely and replaced several stimuli with an insufficient resolution by different photos of the same genus.

In order to assess the number of participants needed for this study, we computed a power analysis using G*Power 3.1 [52] (available at tiny.cc/gpower3). The input parameters were set to a medium effect size (f = 0.15), *p* = 0.0166 (we adjusted the α error probability for multiple comparisons among the categories) and corrected for a correlation among repeated measures computed from the pilot data (r = 0.25). In order to provide a sufficient sample size for a cross-cultural comparison of two nationalities and six different images in a category, the study would require 136 participants.

### 2.4. Participants

A total of 67 Somali and 67 Czech subjects participated in the experiment; however, we had to exclude data of 2 Somali participants due to technical difficulties during the data extraction process. This left 65 Somali and 67 Czech valid measurements. In the Somali sample, there were 24 women and 41 men; the mean age was 22.20 years (range 19–39). In the Czech sample, there were 35 women and 32 men; the mean age was 24.33 years (range 18–44). Most of the participants in both samples were undergraduate students of various fields. No participants expressed extreme attitudes toward the experimental animals—neither positive (e.g., great fondness), nor negative (e.g., strong fear). In the pilot experiment, a total of 31 Czech participants (22 women, 9 men) was tested. None of them participated in the final experiment.

### 2.5. Data Extraction and Curation

The Clinical Eye Tracker measures the position of the participants’ gaze (both monocular and mean binocular) and records approximately 60 samples per second. We developed our own processing software that converts the data into more intuitive variables which were defined as follows. “Number of sample measurements” is the total number of samples measured during the trial (i.e., approx. 300 in our case). A fixation was defined as all sample measurements that are no farther away than 37 pixels (1° visual angle) from a lead (reference) sample measurement. The lead sample measurement was defined as the first measurement, and then each first measurement in the timeline that did not fall inside of the previous fixation. Moreover, each fixation had to consist of at least three consecutive sample measurements (i.e., minimal required duration of one fixation was approx. 50 ms). Following these definitions, we computed the “Number of fixations”. “Fixation time” was defined as the total duration of the participant’s gaze. Finally, “First, second, and third fixation” is the duration of the first, second, and third fixation, respectively. For the purpose of further analyses, we used only mean binocular metrics. Further, we custom-defined three areas of interest (AOI)—left side of the screen, right side of the screen, and fixation cross—and exported all variables separately for each AOI. No AOI overlapped any other.

The data were checked for defected measurements, which were then omitted from the subsequent analyses. The criteria for omitting the measurements were the following: (a) combined fixation time for all AOIs over 5500 ms, and (b) zero number of fixations onto the left or right AOI. The former criterion excluded failed measurements and cases where participants gazed around the AOIs’ borders, leading to assignment of the fixation to multiple AOIs; 19 measurements were excluded based on this criterion. For the latter criterion, we reasoned that the gaze directed toward only one of the stimuli suggests either a lack of interest in the task or extreme emotions toward one of the stimuli, neither of which we wished to measure. In total, 365 from the full number of 4752 measurements were removed from the subsequent analyses. The original data associated with this manuscript are available in Appendix A.

### 2.6. Statistics

All statistical analyses were performed in R [53]. We applied (generalized) linear mixed-effect modelling (nlme package and lme command [54]; and lme4 package and glmer command [55]) to assess whether fear-eliciting stimuli (spiders and scorpions) were fixated upon more often or for a longer time period than the control stimuli (grasshoppers). Firstly, we built a model with an interaction of the position of the focal stimuli (right/left) and the type of the stimuli (pair of animals). The results showed significance for the factors alone, but not their interaction. We therefore computed means for the mirrored pairs of images, and the subsequent analyses were performed on these. When only one of the pairs of stimuli (mirror images) were correctly recorded, we designated this measurement only half the weight of the mean computed from the full pair. Secondly, we constructed a linear mixed-effect model for total fixation time on the stimulus, with animal (stimulus), nationality, gender, and all their interactions as fixed factors and identity of the participant as a random effect. To account for heteroscedasticity, we defined a custom variance structure combining the constant variance structure for participants’ nationality and the fixed variance structure for observation weight. Non-significant fixed effects were removed from the models, and the final model was chosen on the basis of the Akaike information criterion (AIC) and likelihood ratio test. A Tukey post hoc test as implemented in the emmeans package [56] was used to assess pairwise comparisons of retained fixed factors. A similar model was constructed for the total number of fixations. We used a generalized linear mixed-effect model for variables from Poisson distribution (log link) with fixed and random effects as above. Nonetheless, a custom variance structure was not available for this class of models; hence, we only analyzed a subset of the data—only the sum of fixations from full pairs of mirror images (578 out of 4658 measurements were excluded in this analysis). Lastly, we carried out a detailed analysis of the stimuli pairs, i.e., a context-specific analysis of the stimuli. For this analysis, we subtracted fixations on one stimulus in the slide from fixations on the other stimulus in the same slide. This resulted in the difference of the number of fixations on the two stimuli and the difference between fixation times on the two stimuli as responses. Similar to the previous step, we used a pair of stimuli, namely, nationality and gender, and all their interactions as fixed factors, participant identity as a random factor, and the same custom-defined variance structure. The model was reduced accordingly.

## 3. Results

In the linear mixed-effect model for total fixation time, all terms but the three-way interaction (F = 0.60, *p* = 0.547) and the gender–nationality interaction (F = 1.42, *p* = 0.236) proved significant (animal: F = 152.69, *p* < 0.001; nationality: F = 1.37, *p* = 0.244; gender: F = 3.41, *p* = 0.067, animal–nationality interaction: F = 19.18, *p* < 0.001; animal–gender interaction: F = 14.12, *p* < 0.001). The full model vs. final model likelihood ratio was 2.622, *p* = 0.454. For both nationalities and both genders, scorpions were fixated upon for the longest, followed by spiders, while grasshoppers were fixated upon for the shortest time. Nonetheless, there were differences in propensity—Somalis differentiated between the animals to a higher degree than did Czechs, and men differentiated between the animals to a higher degree than did women. For detailed results, see Table 1 and Figure 2.

Models for the total number of fixations gave qualitatively the same results. The final model again included all fixed terms but the three-way interaction and gender–nationality interaction. The full model vs. final model likelihood ratio was 3.511, *p* = 0.319. After further reduction of fixed terms, the reduced models were significantly worse. F-values for fixed effects were as follows: animal: F = 632.42; nationality: F = 13.51; gender: F = 4.95, animal–nationality interaction: F = 82.65; animal–gender interaction: F = 36.19. For both nationalities and both genders, scorpions were fixated upon the most frequently, followed by spiders, while grasshoppers were fixated upon the least frequently (comparisons between levels of animal factors within both nationalities and genders were all highly significant with *p*-value < 0.001). For detailed results, see Appendix A.

In the full linear mixed-effect model for the difference between fixation time on the two stimuli, none of the interactions proved significant (stimuli pair–gender interaction: F = 1.59; *p* = 0.203; stimuli pair–nationality interaction: F = 1.37, *p* = 0.255; nationality–gender interaction: F = 0.24, *p* = 0.625; three-way interaction: F = 0.39, *p* = 0.677); however, all original parameters did. The final model revealed a significant influence of the pair of stimuli (F = 17.77, *p* < 0.001), participant’s gender (F = 6.19, *p* = 0.014), and participant’s nationality (F = 5.91, *p* = 0.016). The full model vs. final model likelihood ratio was 6.967, *p* = 0.432.

Generally, the former animal in the pair (as depicted in Table 2 and Figure 3) attracted more attention than the latter, i.e., both spiders and scorpions were gazed on for a longer time than grasshoppers, and scorpions were gazed on for a longer time than spiders. This difference was higher in men than women; the estimated difference was 295.39 ms (95% confidence interval = 30.40–560.37). Further, it was higher in Somalis than in Czechs; the estimated difference was 322.15 ms (95% confidence interval = 60.00–584.31). For detailed results, see Table 2 and Figure 3.

Similarly, results of the linear mixed-effect models revealed the significant influence of the pair of stimuli (F = 33.40, *p* < 0.001), participant’s gender (F = 5.88, *p* = 0.017), and participant’s nationality (F = 5.03, *p* = 0.027). None of the interactions proved significant (stimuli pair–gender interaction: F = 1.49; *p* = 0.226; stimuli pair–nationality interaction: F = 1.12, *p* = 0.326; nationality–gender interaction: F = 0.11, *p* = 0.737; three-way interaction: F = 0.22, *p* = 0.803). Full model vs. final model likelihood ratio was 5.789, *p* = 0.565.

This model’s results corresponded well with the results of the previous ones. Again, the former animal in the pair attracted more attention than the latter, i.e., participants fixated upon both spiders and scorpions more often than grasshoppers, and scorpions more often than spiders. This difference was higher in men than in women, the estimated difference was 1.26 (95% confidence interval = 0.23–2.28). The estimated difference between Somalis and Czechs was 1.00, Somalis having higher values (95% confidence interval = 0.01–2.02). For detailed results, see Table 3 and Figure 4.

## 4. Discussion

In a clinical and sub-clinical phobic population as well as in healthy controls, emotional stimuli are known to affect the gaze and sway participants’ attention. However, a debate goes on whether these attentional biases are caused by the exceptional ability of the emotional stimuli to attract attention or by the difficulty of disengaging from threat-related stimuli [8,57]. Nonetheless, these biases encompass faster fixation, higher overall number of fixations, and longer fixation durations on the threatening stimuli [43,44,45,58,59]. Likewise, our results show a higher number of fixations (Table 3, Figure 4) and a longer duration of the gaze (Table 2, Figure 3) on the threatening stimuli (spiders and scorpions). This result, therefore, corroborates the previous findings suggesting that the gaze of the participants is directed more on the emotional and fear-eliciting stimuli. Building on these results, the current study adds a neglected but important stimulus—a scorpion—and focuses on a cross-cultural comparison.

Three factors affected the participants’ gaze: gender, pair of stimuli, and nationality. Firstly, women divided their attention more evenly between both images, resulting in a smaller difference of the number of fixations and fixation times for every pair of stimuli. We attribute this to a general gender difference in how we perceive our surroundings. Women tend to spread their attention between various objects and landmarks in the environment, whereas men tend to inspect fewer parts of the environment, yet more thoroughly [60,61,62]. Secondly, we detected a strong effect of the stimuli pair on participants’ gazes. When presented with one non-threatening stimulus (a grasshopper) and one threatening stimulus (a spider or a scorpion), Czech and Somali participants alike directed their attention toward the threatening stimuli. This supported our first hypothesis that chelicerates, in general, are viewed as threatening stimuli compared to similarly looking insects. Moreover, this view is likely shared among the cultures. When presented with two threatening stimuli, a spider and a scorpion, Somalis directed more of their attention toward the scorpion. The same attentional bias in direction, but less pronounced, was also found in Czechs, supporting our second hypothesis. This suggests that scorpions are perceived as a more salient stimulus than the spider in the Somali and to a lesser extent also in the Czech population. Finally, to summarize the effect of nationality, the same attentional bias can be found in both populations, although it is less pronounced in Czechs. Although in terms of the combined effects of gender and nationality some partial comparisons within the model were non-significant (namely spider–grasshopper and scorpion–spider pairs in Czech women), crucially, neither model supported the significance of any interaction. The simple prediction of the scorpion being the most salient, followed by the spider and then the grasshopper, seems robust enough to apply to both cultures and genders.

Cross-cultural comparison is a useful method for testing an evolutionary-based hypothesis. In this particular case, we emphasize the importance of Somalis regarding their continuous presence in the savanna environment of early humans. Even though not a hunter–gatherer society, they still share some traits that might significantly affect human–animal relationships: a semi-nomadic lifestyle, and inhabitation of a diverse, little-artificially cultivated landscape. Under such conditions, recognizing potentially deadly scorpions among other arthropods might have been beneficial for human ancestors. However, only a quick glance is often available before it is necessary to “make a decision”; therefore, we can expect some level of generalization. Similar in body size, body shape, and characteristic mode of motion, the generalization between scorpions and spiders (and probably other arachnids [36]) seems well-based. Ultimately, the cost of misidentifying scorpions as harmless must have outweighed the cost of misidentifying spiders as dangerous. Through natural selection, this trait would then become fixed. In Somalis, the acquired adaptations are reinforced by the continuous presence of scorpions in the evolutionary and physical environment of the Somali population. The secondary stimulus can still activate the same neural pathways, and in the absence of the original stimulus, spiders might become relatively more salient while the saliency of scorpions declines. The previous studies already hypothesized about the generalization of fear among spiders and scorpions [36,37] but were not able to disentangle the direction of generalization; fear of spiders might have originated in fear of scorpions, or fear of spiders might have been generalized on fear of scorpions. Our study addresses this issue. The strong attentional bias towards scorpions in Somalis and to a lesser extent in Czechs corroborates the scorpion as the original threatening stimulus in the generalization of fear among arachnids.

In this study, we suggested an evolutionary explanation for an attentional bias towards scorpions. An obvious objection would be that it is caused simply by the frequency of encountering the animals. Nevertheless, in such a case, we should expect equal attention paid to spiders and scorpions in Somalis (as both are encountered) and a strong attentional bias towards spiders in Czechs (as spiders are encountered on a daily basis while scorpions are completely absent). Neither of these predictions were supported by our data.

Based on Seligman’s [4] influential work on biological preparedness, snakes and spiders are widely investigated as some of the most evolutionary fear-relevant animals (reviewed in [63]). While several lines of evidence seem to support the existence of a series of adaptations to detect, recognize, and avoid snakes (but see [64,65]), results regarding spiders are mixed [5,17,25,27,45]. For example, in an inattentional blindness study, 53% of participants detected, located, and identified a spider as opposed to only 10% of participants who were able to detect, locate, and identify a housefly, suggesting prioritized attention toward spiders [5]. In a different study, both spiders and snakes were detected faster than mushrooms among fruit distractors, but snakes were detected even faster than spiders in attentionally more challenging conditions (more distractors or shorter exposure to the stimuli matrix), suggesting higher sensitivity of spider attentional bias to “ecological” conditions [17]. It would be compelling to replicate these experiments with scorpions among the tested stimuli. Such experiments, however, must be conducted on a general population. Although it is tempting to view arachnophobia (the specific phobia of spiders) as a simple extension of a high fear of spiders, in some contexts, it is rather misleading. For example, potential arachnophobes ranked spiders similarly to respondents with only high or medium fear of spiders, differing only in the absolute intensity of negative emotions. Nevertheless, unlike the general population, they feared almost all spiders more than any other stimulus [38]. In a visual search task, potential arachnophobes found spiders among distractors significantly faster than snakes (a subjectively non-feared but evolutionary fear-relevant animal [44,66]). In other words, although a deep fear of scorpions, a deep fear of spiders, and arachnophobia might share evolutionary roots and even largely utilize the same neural circuits, arachnophobia is simply a specific phobia. Hence, we do not expect arachnophobes to be triggered by scorpions as much as they are by spiders. To summarize, the spider–scorpion generalization hypothesis attempts to explain the roots of strong negative emotions associated with spiders in (Western) societies, as well as discrepancies in evidence obtained under the paradigm of spiders as an evolutionary relevant threat by offering a slightly altered scenario where scorpions represent the evolutionary primary stimulus.

Nonetheless, different hypotheses have also been considered in the last decades. Cultural influence, learning, or disease-avoidance model are a few explanations for the deep fear of spiders that have been suggested as an alternative to or an extension of the biological preparedness and related theories [34,35,67,68]. More research is needed on this topic. For example, the apparent lack of scorpion phobia might be considered an argument against the spider–scorpion generalization hypothesis. After all, it would be rather puzzling that there is a phobia from the evolutionary secondary stimulus but not the primary one. However, as discussed in Frynta et al. [36], this might be the result of the WEIRD countries’ research bias, patient report bias, and collective labelling of animal phobias (for example, an impressive cross-national study of phobias was conducted by Wardenaar et al. [69], but all animal phobias were assessed and reported as one). We expect the scorpion phobia to be at least as prevalent as the spider phobia in certain regions of the world, for example in the Middle East or rural areas of the southwestern USA. During data collection for this research, we encountered one Somali man who literally jumped off his chair when we explained he would see images of scorpions (but he expressed no problem with watching spiders). We hope this anecdote may inspire researchers based in the above-mentioned regions to investigate whether something like a scorpion phobia really exists.

## 5. Conclusions

While the evolutionary origin of the fear of spiders remains uncertain, this study supports the idea of it being rooted in a generalized fear of chelicerates. The prioritized attention towards deadly scorpions would have been advantageous for our human ancestors, and, due to their similarity in appearance, spiders might elicit a similar reaction utilizing the same neural circuits. In Somalis, the acquired adaptations would be reinforced by scorpions’ continuous presence in the environment, whereas in the scorpions’ absence, spiders might become relatively more salient, as was shown in the Czechs. The spider–scorpion generalization hypothesis represents a modified scenario of the classical biological preparedness theory applied to the evolutionary origin of the fear of spiders. As such, it offers possible explanations for some contradictory results, for example, different attitudes toward spiders around the world or spiders’ intermediate saliency in visual search tasks. Lastly, this study adds to a very small pool of studies investigating people’s perception of animals in Sub-Saharan Africa.

## Figures and Tables

**Figure 1 animals-12-03466-f001:**
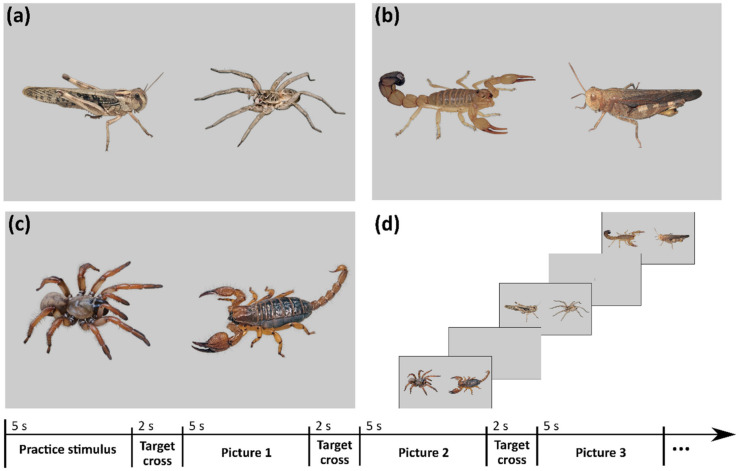
Examples of experimental stimuli, slide presentation, and timeline of the experiment. Pairs of stimuli were of three types: (**a**) spider–grasshopper, (**b**) scorpion–grasshopper, and (**c**) spider–scorpion. (**d**) During the experiment, pictures with stimuli were interlaced by 20% grey slides with a target cross in the middle. The axis shows a timeline of the experiment; altogether, there were 36 pictures with stimuli pairs. The species are as follows, from left to right: (**a**) migratory locust (*Locusta migratoria*), photo by Daniel Frynta, and a wolf spider *Lycosa leuckarti*, photo by Bidgee via Wikimedia Commons (CC BY-SA 3.0); (**b**) fattail scorpion (*Androctonus australis*), photo by Daniel Frynta, and sulphur-winged grasshopper (*Arphia sulphurea*), photo by Judy Gallagher via Flickr (CC BY 2.0); (**c**) North American mygalomorph spider of the genus *Aptostichus*, photo by Marshal Hedin via Flickr (CC BY-SA 2.0), and the scorpion *Pandinurus phillipsii*, photo by Daniel Frynta.

**Figure 2 animals-12-03466-f002:**
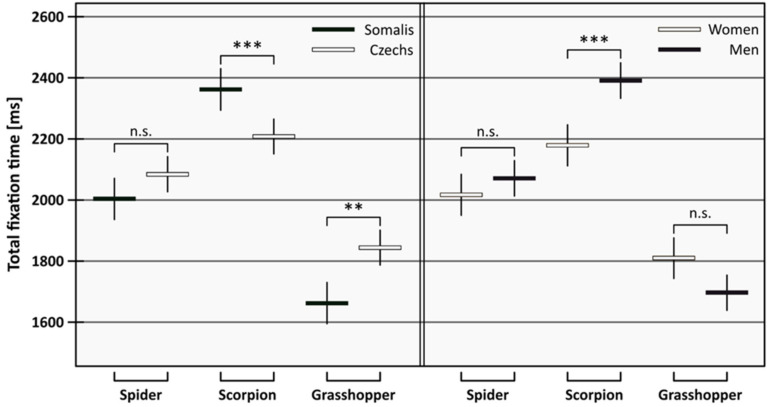
Total fixation time on the animal stimuli for Somalis and Czechs (left panel) and women and men (right panel). Error bars are 95% confidence intervals. Means are tested using Tukey post hoc test, significances are indicated by asterisks (n.s. *p* ≥ 0.05; ** *p* < 0.01; *** *p* < 0.001).

**Figure 3 animals-12-03466-f003:**
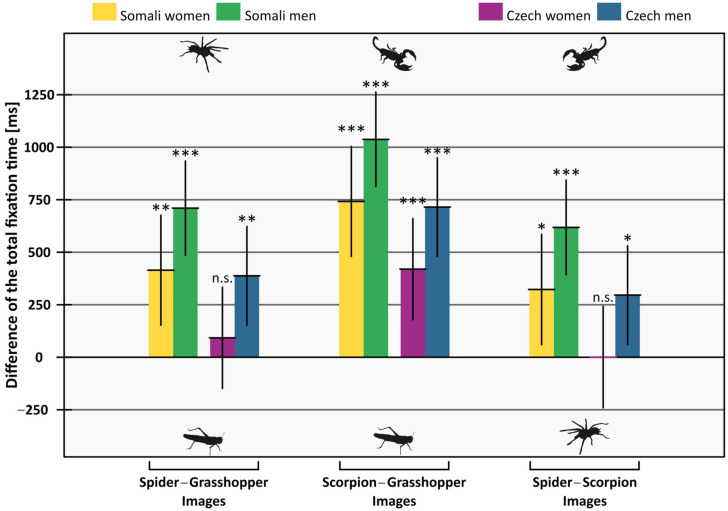
Mean difference of the total fixation time on the two stimuli. When the difference is positive, participants looked more on the first stimuli of the pair (e.g., in Spider−Grasshopper Images, participants looked more on the spiders). Error bars are 95% confidence intervals. Means are tested against zero, significances are indicated by asterisks (n.s. *p* ≥ 0.05; * *p* < 0.05; ** *p* < 0.01; *** *p* < 0.001).

**Figure 4 animals-12-03466-f004:**
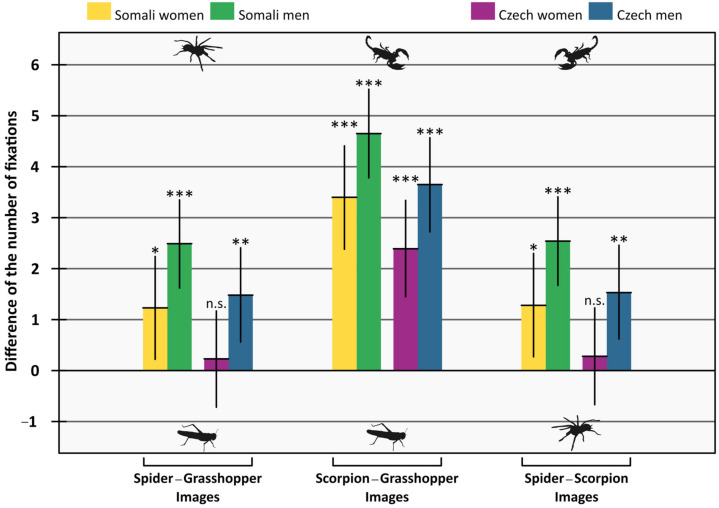
Mean difference of the number of fixations on the two stimuli. When the difference is positive, participants looked more on the first stimuli of the pair (e.g., in Spider−Grasshopper Images, participants looked more on the spiders). Error bars are 95% confidence intervals. Means are tested against zero, and significances are indicated by asterisks (n.s. *p* ≥ 0.05; * *p* < 0.05; ** *p* < 0.01; *** *p* < 0.001).

**Table 1 animals-12-03466-t001:** Results of the linear mixed effect model for the total fixation time. Upper part—the estimated means and 95% confidence intervals (in milliseconds) for each stimulus (animal) within both nationalities and genders. Lower part—comparison of animal factor levels for both nationalities and genders; Est.—estimated difference; t-rat.—t-ratio; *p*-values < 0.05 are in bold.

Animal	Somalis			Czechs			Women			Men		
	Mean	95% CI		Mean	95% CI		Mean	95% CI		Mean	95% CI	
Spider	2004	1936, 2071		2084	2027, 2142		2017	1950, 2084		2071	2013, 2129	
Scorpion	2362	2294, 2430		2208	2151, 2265		2179	2112, 2246		2391	2333, 2449	
Grasshopper	1662	1595, 1730		1844	1787, 1901		1810	1743, 1876		1697	1639, 1754	
**Contrast**	**Est.**	**t-rat.**	** *p* **	**Est.**	**t-rat.**	** *p* **	**Est.**	**t-rat.**	** *p* **	**Est.**	**t-rat.**	** *p* **
Spider–Grasshopper	341	7.25	**<0.001**	240	6.07	**<0.001**	207	4.47	**<0.001**	375	9.32	**<0.001**
Scorpion–Grasshopper	700	14.82	**<0.001**	364	9.22	**<0.001**	370	7.96	**<0.001**	694	17.27	**<0.001**
Scorpion–Spider	358	7.57	**<0.001**	124	3.12	**0.022**	163	3.49	**0.006**	319	7.93	**<0.001**

**Table 2 animals-12-03466-t002:** Results of the linear mixed effect models for the difference between fixation time on the two stimuli−the estimated means for all levels of stimuli pair, nationality, and gender. All predicted estimates are tested against zero−positive values signify longer fixation time on the former animal in the pair and negative values on the later animal in the pair; *p*-values < 0.05 are in bold.

	Pair of Stimuli	Somalis				Czechs			
		Mean	95% CI	t	*p*	Mean	95% CI	t	*p*
**Women**	Spider–Grasshopper	414.54	152.92, 676.16	3.12	**0.002**	92.39	−148.74, 333.51	0.75	0.453
	Scorpion–Grasshopper	741.86	479.76, 1003.96	5.55	**<0.001**	419.71	178.75, 660.67	3.42	**<0.001**
	Scorpion–Spider	322.81	60.24, 585.38	2.41	**0.016**	0.66	−240.72, 242.04	0.01	0.996
**Men**	Spider–Grasshopper	709.93	485.04, 934.81	6.19	**<0.001**	387.77	152.63, 622.91	3.23	**0.001**
	Scorpion–Grasshopper	1037.25	812.02, 1262.48	9.03	**<0.001**	715.09	480.33, 949.86	5.97	**<0.001**
	Scorpion–Spider	618.20	392.60, 843.79	5.37	**<0.001**	296.04	61.02, 531.07	2.47	**0.014**

**Table 3 animals-12-03466-t003:** Results of the linear mixed effect models for the difference of the number of fixations on the two stimuli−the estimated means for all levels of stimuli pair, nationality, and gender. The predicted estimates are tested against zero−positive values signify more fixations on the former animal in the pair and negative values on the later animal in the pair; *p*-values < 0.05 are in bold.

	Pair of Stimuli	Somalis				Czechs			
		Mean	95% CI	t	*p*	Mean	95% CI	t	*p*
**Women**	Spider–Grasshopper	1.23	0.22, 2.24	2.39	**0.017**	0.23	−0.72, 1.17	0.47	0.639
	Scorpion–Grasshopper	3.40	2.38, 4.41	6.58	**<0.001**	2.39	1.45, 3.34	4.96	**<0.001**
	Scorpion–Spider	1.28	0.27, 2.30	2.48	**0.013**	0.28	−0.67, 1.23	0.58	0.563
**Men**	Spider–Grasshopper	2.49	1.62, 3.35	5.62	**<0.001**	1.48	0.56, 2.41	3.15	**0** **.002**
	Scorpion–Grasshopper	4.65	3.78, 5.52	10.49	**<0.001**	3.65	2.72, 4.57	7.75	**<0.001**
	Scorpion–Spider	2.54	1.67, 3.41	5.72	**<0.001**	1.53	0.62, 2.46	3.26	**0** **.001**

## Data Availability

The original data associated with this manuscript are available in the Appendix A. Requests for uncurated data should be addressed to P.F.

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
