# Peer review of "Do Spiders Ride on the Fear of Scorpions? A Cross-Cultural Eye Tracking Study"

_animals, 2022, doi:10.3390/ani12243466_

Round 1

Reviewer 1 Report

Dear Editor,

First, I should admit that I enjoyed reading this manuscript. The objective and general hypotheses are easy to understand. Moreover, the main hypothesis is refreshing and exciting. Also, the methods - except for the statistical analyses - are very well presented. In the general discussion, the authors presented the limits of their study and explored different alternatives, which I appreciated.

Unfortunately, I consider that the data (and statistical analyses) do not support the main conclusion, which suggests a significant difference between the two cultures (Czechs and Somalians) regarding the level of « fear » elicited by the stimuli (spiders, scorpions, grasshoppers). The statistical analyses are problematic, and I'll concentrate my review on this particular element.

In conclusion, I recommend reconsidering the manuscript after major revision (especially regarding the presentation of the results, the statistical analyses and the main conclusion). If the authors resubmit a manuscript presenting a new conclusion (supported by the data and the statistical analyses), I would be delighted to read this manuscript once revised.

Statistical analyses

The authors analyzed the number of fixations and the duration of fixation. Therefore, I will examine both variables separately. 

(1) Linear mixed effect model for the difference in the number of fixations

Technically, the number of fixations (a frequency) is a discrete measure and should be evaluated using a Poisson (or negative binomial) distribution - not a Gaussian distribution, as performed in the current manuscript. However, the authors complicated things by attributing a half-frequency for this measure, making it « numerical» . Honestly, I do not know how to treat a half-frequency because it is not discrete (but not numerical).

On page 6, the authors wrote, "Non-significant fixed effects were removed from the models, and the final model was chosen on the basis of Akaike information criterion (AIC). » I do not understand this sentence because the final models included all possible interactions. Also, the authors should use a likelihood ratio test to compare the models, not just the AIC values.

Given that the interaction was not significant, I do not see why the authors presented and tested all possible interactions between nationalities, gender and stimuli (see Table 1 and Figure 2).

Also, given that the "model revealed significant influence of the participant's gender (F = 5.88, p = 0.017) » [see p.6], it is irrelevant to present a t-test showing the same result two lines below. See «  This difference was higher in men than women, the estimated difference was 1.26 (95% confidence interval = 0.23 – 2.28, t = 2.42, p = 0.017). » 

Similarly, the "model revealed significant influence of the participant's nationality (F = 5.03, p = 0.027) » [see p.6]. But a few lines later, the authors present a two-tailed t-test testing the same thing ("The estimated difference between Somalis and Czechs was 1.00, Somalis having higher values (95% confidence interval = 0.01 – 2.02, t = 1.96, p = 0.052). "). 

Given the absence of interaction, Table 1 and Figure 2 are irrelevant and should be removed from the manuscript. Thus, the authors should interpret the main effects (pairs of stimuli, gender, nationality) by comparing the means of each level of these effects. No additional analysis is necessary here.

Also, the authors wrote, «  Generally, the former animal in the pair attracted more attention than the latter, i.e., participants fixated both spiders and scorpions more often than grasshoppers, and scorpions more often than spiders. »  However, the authors do not present a posteriori tests [comparing spiders, scorpions and grasshoppers solely]  to support this conclusion.

(2) Linear mixed effect models for the difference between fixation time on the two stimuli 

Overall, the authors duplicated the same problems for the variable fixation time. 

Once again, the authors rerun a two-tailed test on the main effects (pairs of stimuli, gender and nationality), duplicating the analyses.

Given the absence of interaction, Table 2 and Figure 3 are irrelevant and should be removed from the manuscript. Thus, the authors should interpret the main effects (pairs of stimuli, gender, nationality) by comparing the means of each level of these effects. No additional analysis is necessary here.

Also, the authors wrote, «  Generally, the former animal in the pair attracted more attention than the latter, i.e., spiders and scorpions were gazed on for a longer time than grasshoppers, and scorpions were gazed on for a longer time than spiders. ». However, the authors do not present a posteriori tests [comparing spiders, scorpions and grasshoppers solely]  to support this conclusion.

Discussion

In short, this study reveals that the participants spent more time looking at one of the stimuli (possibly the scorpions - to be supported by a posteriori tests), that males looked longer at the images than females and that Somali participants looked longer at the images than Czechs. 

Unfortunately, the results and statistical analyses presented in the manuscript do not support the main conclusion advanced by the authors, which is that "Czech and Somali participants alike directed their attention toward the threatening stimuli. » My reading of the manuscript does not support this conclusion.

Additional comments

In the future, it would be nice to add a physiological measure to the participants' fear response. This way, the authors could correlate the looking frequency (or time) with the physiological fear response of the participants.

By fading the colour of the spiders, scorpions and grasshoppers, did the authors reduce the participants' fear response?

Conclusion

The authors need to remove the t-tests they performed after the main analyses. In addition, the authors need to remove Tables 1 and 2 and Figures 2 and 3, which suggest an interaction between the factors - but there was no interaction. The authors also need to change the main interpretation of the manuscript, which, in its current form, suggests a difference between Czech and Somali participants relative to each type of stimuli (particularly spiders and scorpions). Finally, they must rewrite the discussion to consider only the main effects of their analyses, which, unfortunately, do not support the principal hypothesis of the study.

Reviewer 2 Report

Dear Authors

Your manuscript is well-written and describes research of great interest to the readers of the Animals journal. I will recommend your publication to the editor.

There are some changes and additions that I recommend to you, in the list below.

1) the title is unclear and does not tell readers what the article is about. I recommend that you replace it with a newspaper headline type title - very direct and informative. Something like, for example, "A Crosscultural Eye Tracking Study supports the hypothesis that fear of spiders originated as a generalized fear of scorpions"

2) Please include more keywords in the manuscript. They make the article more visible on search engines like Google Scholar, and they also help readers decide to read it.

3) The Introduction is well written. However, it would be advisable to include in that section at least one paragraph explaining the concepts of biophilia and biophobia, from an evolutionary perspective. This addition would broaden readers' understanding of the context of the research developed by the authors. See, for example, the paper Patuano, Agnès. "Biophobia and urban restorativeness." Sustainability 12.10 (2020): 4312.

4) I strongly recommend that figures 2 and 3 be modified: the comparative bars between Somali men/women and Czech men/women should contain contrasting colors for better visualization.

Greetings from Brazil,

The Reviewer

Round 2

Reviewer 1 Report

Dear editor,

The authors presented a detailed response regarding my concerns about the statistical analyses of this study. I am pleased with the modifications, and I confirm that the results section aligns with the study's conclusions. Therefore, I support the publication of this manuscript in Animals.

Sylvain Fiset